# Deep Learning Solution of the Eigenvalue Problem for Differential Operators

## Abstract

Solving the eigenvalue problem for differential operators is a common problem in many scientific fields. Classical numerical methods rely on intricate domain discretization, and yield non-analytic or non-smooth approximations. We introduce a novel Neural Network (NN)-based solver for the eigenvalue problem of differential self-adjoint operators where the eigenpairs are learned in an unsupervised end-to-end fashion. We propose three different training procedures, for solving increasingly challenging tasks towards the general eigenvalue problem. The proposed solver is able to find the $M$ smallest eigenpairs for a general differential operator. We demonstrate the method on the Laplacian operator which is of particular interest in image processing, computer vision, shape analysis among many other applications. Unlike other numerical methods such as finite differences, the partial derivatives of the network approximation of the eigenfunction can be analytically calculated to any order. Therefore, the proposed framework enables the solution of higher order operators and on free shape domain or even on a manifold. Non-linear operators can be investigated by this approach as well.

## 1 Introduction

Eigenfunctions and eigenvalues of the Laplacian (among other operators) are important in various applications ranging, inter alia, from image processing to computer vision, shape analysis and quantum mechanics. It is also of major importance in various engineering applications where resonance is crucial for design and safety [Benouhiba & Belyacine (2013)]. Laplacian eigenfunctions allow us to perform spectral analysis of data measured at more general domains or even on graphs and networks [Shi & Malik (2000)]. Additionally, the $M$-smallest eigenvalues of the Laplace-Beltrami operator are fundamental features for comparing geometric objects such as 3D shapes, images or point clouds via the functional maps method in statistical shape analysis [Ovsjanikov et al. (2012)]. Moreover, in quantum mechanics, the smallest eigenvalues and eigenfunction of the Hamiltonian are of great physical significance [Han et al. (2019)]. In this paper we present a novel numerical method for the computation of these eigenfunctions (efs) and eigenvalues (evs), where the efs are parameterized by NNs with continuous activation functions, and the evs are directly calculated via the Rayleigh quotient. The resulting efs are therefore *smooth* functions defined in a parametric way. This is in contrast to the finite element [Pradhan & Chakraverty (2019)] and finite difference [Saad (2005); Knyazev (2000)] methods in which the efs are defined on either a grid or as piecewise linear/polynomial functions with limited smoothness. In these matrix-based approaches one has to discretize first the problem and to represent it as an eigenvalue problem for a matrix. This in itself is prone to numerical errors.

Following [Bar & Sochen (2019)], we suggest an unsupervised approach to learn the eigenpairs of a differential operator on a specified domain with boundary conditions, where the network simultaneously approximates the eigenfunctions at every entry $x$. The method is based on a uniformly distributed point set which is trained to satisfy two fidelity terms of the eigenvalue problem formulated as the $L_2$ and $L_\infty$-like norms, boundary conditions, orthogonality constraint and regularization. There are several advantages of the proposed setting: (i) the framework is general in the sense that it can be used for non linear differential operators with high order derivatives as well. (ii) Since we sample the domain with a point cloud, we are not limited to standard domains. The problem can be therefore solved in an arbitrary regular domain. (iii) The framework is generic such that additional constraints and regularizers can be naturally integrated in the cost function. (iv) Unlike

previous methods, the suggested framework solves *simultaneously* multiple eigenpairs. This means that we handle a family of PDEs (one for each eigenvalue and finding the eigenvalues themselves) in one network that solves these multiple PDEs together. The method is applied in 1D and 2D for both known and multiple unknown eigenvalues of the Laplacian operator. Quantitative analysis demonstrates the robustness of the method compared with the classical matrix-based methods.

## 2 RELATED WORK

Many recent approaches have shown promise in using the power of NNs to approximate solutions of differential equations. Classical methods are often prone to weakness due to the discretization of the domain $\Omega$. In [Bar & Sochen (2019)], the authors propose a solver for both forward and inverse problems, using NNs to model the solution, and penalizing using both the automatic differentiation, and boundary conditions. In [Raissi et al. (2017)], a similar approach was taken to solve both continuous and discrete time models. In [Chen et al. (2018)], differential equation solvers are used as part of the network architecture, and are shown to enhance the smoothness and convergence of the solutions. In order to properly solve differential equations, a representation that captures high-order derivatives is desired. Recently, [Sitzmann et al. (2020)] proposed a network architecture that illustrates these requirements using periodic activation functions with the proper initialization. Additionally, [Rippel et al. (2015)] proposed leveraging Discrete Fourier Transform (DFT) to represent the network in spectral space.

The well-known power method and its variants [Eastman & Estep (2007)] has been the main method for addressing the eigenvalue problem. The method works on specific *Linear* operators, $\mathcal{L} : L_2(\mathbb{R}^d) \to L_2(\mathbb{R}^d)$. It is done after the continuous equation is reduced numerically to an eigen-pair equation for matrices. This process introduces numerical errors even before the solution of the eigen problem. The usage of the power method for spectral operators on Hilbert Spaces was recently shown inn [Erickson et al. (1995)]. In [Hait-Fraenkel & Gilboa (2019)] a modified method for non-linear differential operators was proposed. Furthermore, most power method variants for operators, converge to a single eigenpair. Finding the $M$ smallest eigenpairs can be both computationally and algorithmically challenging.

Advanced methods addressing the eigenvalue problem via deep networks were recently introduced. These methods are based on variational Monte Carlo (VMC) and diffusion Monte Carlo (DMC) methods. VMC relies on leveraging physical knowledge to propose an ansatz of the eigenfunction and incorporates the essential physics [Han et al. (2019); Hermann et al. (2019); Pfau et al. (2019); Choo et al. (2019)]. Recently, [Han et al. (2020)] formulated the eigenvalue problem by the stochastic backward equation using the DMC method, where the loss function optimizes the eigenvalue, eigenfunction and the scaled gradient of the eigenfunction. The loss function consists of $L_2$ norm of two fidelity terms with additional normalization. The algorithm yields the first eigenpair with an optional second eigenpair given some mild prior estimate of the eigenvalue. In the suggested work, we formulate the eigenvalue problem in a direct setting with flexible number of eigenpairs. Additionally, we use $L_\infty$ norms for fidelity and boundary condition terms to accomplish a strong (pointwise) solution.

## 3 PRELIMINARIES

Let $\mathcal{H}$ be a Hilbert space where the inner product for $u, v \in \mathcal{H}$ is $\langle u, v \rangle$. Let $A \in O(\mathcal{H})$ be an operator. Let $A^*$ be the adjoined operator defined by $\langle A^* u, v \rangle = \langle u, Av \rangle \ \forall u, v \in \mathcal{H}$. Then $A$ is said to be self-adjoint if $A = A^*$. We start with a short Lemma on self-adjoint operators [Conway (1985)].

**Lemma 3.1** *Let $\mathcal{H}$ be a Hilbert space. Let $A \in O(\mathcal{H})$ be a self-adjoint operator. Then all eigenvalues of $A$ are real.*

In this work we focus on self-adjoint operators. An eigenpair of an operator $\mathcal{L}$ is defined as: $(u, \lambda)$ s.t. $\lambda \in \mathbb{R}$, where $u$ is the eigenfunction of $\mathcal{L}$ and $\lambda$ is the corresponding eigenvalue. Let $\mathcal{L}$ be a self-adjoint operator $\mathcal{L} : L_2(\mathbb{R}^d) \to L_2(\mathbb{R}^k)$. Our objective is to search for eigenpairs $\{u_i, \lambda_i\}$ such that

$$\mathcal{L}u_i + \lambda_i u_i = 0 \ \forall i. \tag{1}$$

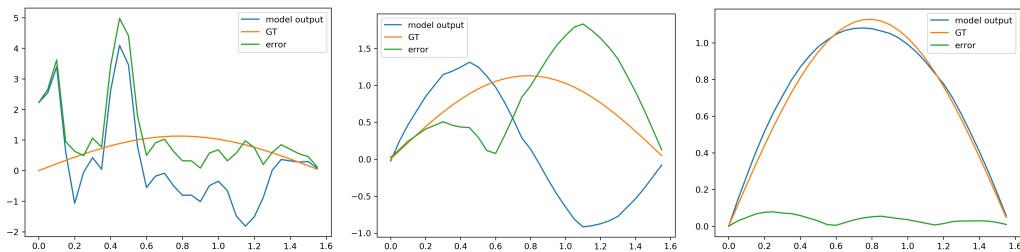

Figure 1: The solution to $u'' + 4u = 0$ with $u(0) = u(\pi/2) = 0$ at iterations (from left to right) 1, 100 and 2500. At the last iteration RMSE = 6.52e−4 and PSNR = 28.84.

The proposed algorithm approximates the eigenfunction set $u_i(x)$ by a NN $u_i(x; \theta_{u_i})$ parameterized by $\theta_{u_i}$ which denotes the network weights and biases. The network consists of few fully connected layers with smooth activation function $\varphi$ and linear sum in the last layer. For example, four layers architecture is given by

$$u(x) = W_5\varphi\Big(W_4\varphi\Big(W_3\varphi\Big(W_2\varphi\Big(W_1x + b_1\Big) + b_2\Big) + b_3\Big) + b_4\Big) + b_5, \qquad (2)$$

and $\varphi(\cdot) = \tanh(\cdot)$ or SIREN. The input to the network is $x \in \mathbb{R}^d$, one input node for each dimension. The network is trained to satisfy the PDE with boundary conditions by minimizing a cost function.

## 4 SINGLE KNOWN EIGENVALUE

We first address the problem of finding a single eigenfunction $u(x)$ given its corresponding eigenvalue $\lambda$. We approximate $u(x)$ by a NN and optimize the following cost function

$$\mathcal{F}_1\left(u(x, \theta_u)\right) = \alpha\|\mathcal{T}u\|_2^2 + \mu\|\mathcal{T}u\|_\infty + \delta\|u - u_0\|_{1,\partial\Omega} + \beta\Big|\|u\|_2^2 - c\Big| + \rho\|\theta_u\|_2^2, \qquad (3)$$

where

$$\mathcal{T}u := \mathcal{L}u + \lambda u,$$

with the Laplacian operator $\mathcal{L} = \Delta$. The first two terms in equation 3 are the $L_2$ and $L_\infty$ fidelity terms where the latter promotes a *pointwise* solution such that the equation is satisfied for isolated points as well [Bar & Sochen (2019)]. The third term imposes boundary conditions and the forth normalizes the squared length of $u$ to $c = 1$ due to scale invariance of the eigenfunction (since $(\alpha u, \lambda)$ is a valid eigenpair for $\alpha \neq 0$). The final term is the standard weight decay term which stabilizes the network weights. The $L_2$ norm $\|.\|_2^2$ is defined as the Monte-Carlo Integral approximation norm:

$$\|u\|_2^2 = \frac{\text{Vol}(\Omega)}{N} \sum_{x=x_1}^{x_N} |u(x)|^2 \sim \int_\Omega |u(x)|^2 dx. \qquad (4)$$

In the first example we apply the Laplacian operator in 1D where $\lambda = 4$ and $u(0) = u(\pi/2) = 0$. Then,

$$\mathcal{T}u = u''(x) + 4u(x).$$

The normalized analytical solution is therefore

$$u(x) = \frac{2}{\sqrt{\pi}}\sin(2x).$$

Figure 1 demonstrates the outcome of the algorithm at iterations 1, 100 and 2500. As can be seen, the approximated solution approaches the sin function with Relative Mean Square Error (RMSE) = 6.52e−4 and peak signal-to-noise ratio (PSNR) = 28.84.

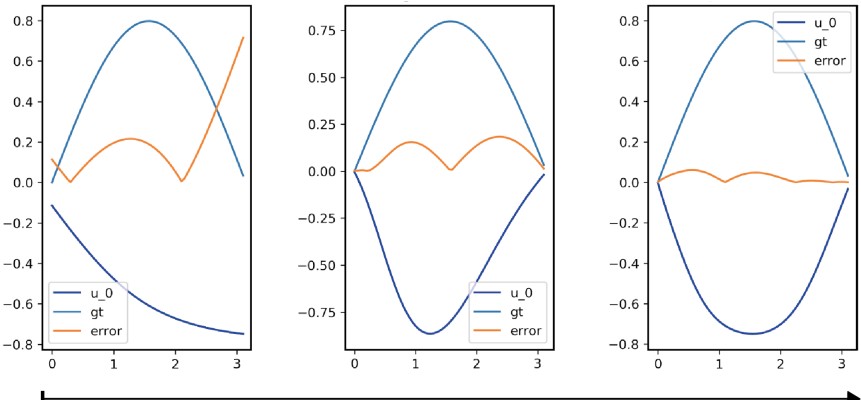

Figure 2: The solution to $u'' + \lambda u = 0$ with $u(0) = u(\pi) = 0$ at iterations 1, 500 and 1000. The associated eigenvalues are $\lambda = 70 \pm 53$, $5.5 \pm 0.5$, $1 \pm 0.03$. Note that the solution is up to a sign.

## 5    SINGLE EIGENPAIR WITH THE SMALLEST EIGENVALUE

Next, we address the case where the eigenvalue is not known in advance. We therefore limit ourselves to the smallest nontrivial eigenpair. This approach is analogue to power method approach, where only one dominant eigenpair is found. Recall the Rayleigh quotient defined as [Miller (2016); Feld et al. (2019)]

$$R(u) := -\frac{\langle \mathcal{L}u, u \rangle}{\langle u, u \rangle}. \tag{5}$$

It can be shown that the eigenfunction $u$ is a critical point of $R(u)$, where $R(u)$ is its corresponding (nontrivial) eigenvalue $\lambda$. Furthermore, if $\tilde{u}$ is a function which is close to $u$ then $R(\tilde{u})$ approximates $\lambda$. In the following cost function we replace $\lambda$ by $R(u)$. Then equation 3 is modified to

$$\mathcal{F}_2(u(x,\theta_u)) = \alpha \|\mathcal{T}u\|_2^2 + \mu \|\mathcal{T}u\|_\infty + \delta \|u - u_0\|_{1,\partial\Omega} + \beta \Big| \|u\|_2^2 - c \Big| + \rho \|\theta_u\|_2^2 + \gamma \|R(u)\|_2^2, \tag{6}$$

where

$$\mathcal{T}u = \mathcal{L}u + R(u)u.$$

The last term minimizes $R(u)$ and therefore attracts the solution to the lowest nontrivial eigenvalue. The ground truth eigenpair is given by $\left( \sqrt{\frac{2}{\pi}} \sin(x), 1 \right)$ for $\Omega = [0, \pi]$. Figure 2 shows the outcome of the proposed method at three iterations. The approximated eigenfunction is obtained up to a sign. The eigenvalue which is the value of the Rayleigh quotient converges to the true value $\lambda = 1$ with decreasing standard deviation along the mini-batches. Quantitative results are shown in row 1 of tables 1 and 2 where MAE stands for Mean Absolute Error and MRE for Mean Relative Error.

## 6    MULTIPLE EIGENPAIRS WITH THE SMALLEST EIGENVALUES

A generalization of the former case is to find $M$ eigenpairs with the corresponding bottom-$M$ eigenvalues. Following [Hait-Fraenkel & Gilboa (2019)], and using the orthogonality property of the eigenfunctions we optimize the following cost function:

$$\mathcal{F}_3(\mathbf{u}, (x, \theta_u)) = \sum_{i=1}^{M} \Big( \alpha \|\mathcal{T}u_i\|_2^2 + \mu \|\mathcal{T}u_i\|_\infty + \delta \|u_i - u_0\|_{1,\partial\Omega} + \beta \Big| \|u_i\|_2^2 - c \Big| + \gamma_i \|R(u_i)\|_2^2 \Big)$$

$$+ \rho \|\theta_u\|_2^2 + \nu \sum_{i<j} \langle u_i, u_j \rangle, \tag{7}$$

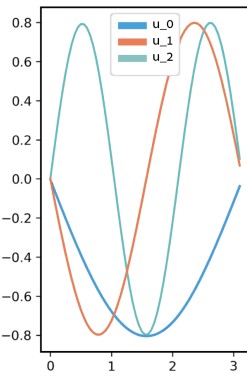 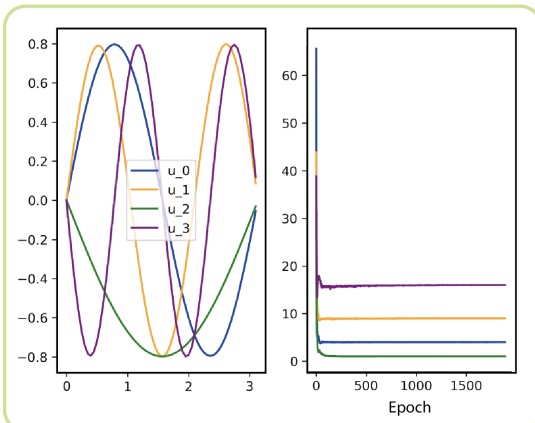

Figure 3: Left panel: eigenfunctions for $M = 3$. Right panel: eigenfunctions for $M = 4$ are shown on the left and the convergence of the eigenvalues are shown on the right. Clearly the convergence of the functions are $u_n(x) = \sqrt{\frac{2}{\pi}}\sin(nx), \; \lambda_n = n^2$, up to a sign.

Table 1: Dimension= 1, Eigenfunction Error Analysis

| $M$ | Mean PSNR (per ef.) | Mean RMSE (per ef.) |
|---|---|---|
| 1 | 28.84 | 1.30e-3 |
| 2 | 56.58 | 2.21e-6 |
| 3 | 52.21 | 8.54e-6 |
| 4 | 48.78 | 2.05e-5 |

where the last term is explicitly given by

$$\nu \sum_{k=1}^{N_s} \sum_{i=1}^{M} \sum_{j=i+1}^{M} u_i(x_k)u_j(x_k),$$

with point set size $N_s$. For each eigenfunction $u_i$ we impose the $L_2$ and $L_\infty$ terms, boundary conditions and normalization as before. The weight of the Rayleigh quotient is multiplied by $\gamma_i = 1/i$ since we want a monotonic penalty of the size of $\lambda_i$. The last term enforces the orthogonality of distinct eigenfunctions. Figure 3 demonstrates the outcome of the algorithm in 1D. In this case we have one network with $M$ output values, one for every eigenfunctuion. The ground truth eigenfunctions of the Laplacian with $u(0) = u(\pi) = 0$ are given by

$$u_n(x) = \sqrt{\frac{2}{\pi}}\sin(nx), \; \lambda_n = n^2, \; n = 1, 2, 3, \dots \quad .$$

The left panel of Figure 3 shows the results for $M = 3$ and in the right for $M = 4$. The right figure in the right panel shows the convergence of the four eigenvalues which as expected have the values 1, 4, 9 and 16. Quantitative results are summarized in tables 1 and 2.

Next, we tested the method in two-dimensions where we trained $M$ different networks simultaneously, each with a single output, one for each eigenfunction. We found this architecture adequate for the multiple eigenpairs problem in the 2D case. The ground truth solution is then

$$u_{nm}(x, y) = \frac{2}{\pi}\sin(nx)\sin(my), \; \lambda_{nm} = n^2 + m^2, \; n, m = 1, 2, 3 \dots$$

for $\Omega = [0, \pi]^2$ and $u(x, 0) = u(x, \pi) = u(0, y) = u(\pi, y) = 0$. Figure 4 shows the results at different iterations for $M = 1$. The expected eigenvalue is the lowest one ($n = 1, m = 1$).

Table 2: Dimension= 1, Eigenvalue Error Analysis

| $M$ | Ground Truth | Predicted | Mean MAE (per ev.) | Mean MRE (per ev.) |
|---|---|---|---|---|
| 1 | 1 | 1.02 | 0.02 | 0.02 |
| 2 | 1, 4 | 1.11, 4.09 | 0.10 | 0.07 |
| 3 | 1, 4, 9 | 1.08, 8.93, 4.13 | 0.09 | 0.04 |
| 4 | 1, 4, 9, 16 | 1.12, 4.09, 9.03, 15.95 | 0.07 | 0.04 |

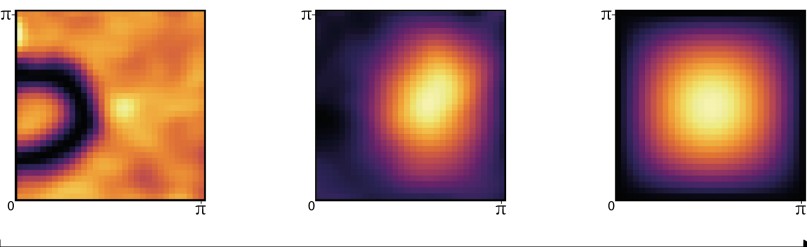

Figure 4: The solution to the 2D Laplace eigevalue problem, with $M = 1$ at iterations (from left to right) 1, 100 and 1000.

As the algorithm converges, the expected eigenfunction is clearly seen (right image). We further tested our performance for $M = 4$, see Figure 5. The figures from left to right stand for the four eigenfunctions. The rows from top to bottom are iterations 1, 500 and 1000 respectively. As can be easily shown, these results are with accordance with the theoretical eigenfunctions $(n, m) = (1, 1), (1, 2), (2, 1), (2, 2)$. Quantitative results are summed up in tables 3 and 4 where SSIM stands for structural similarity index measure.

As the proposed method does not depend on the discretization of the domain, it can be easily adapted to free-form domains. It is demonstrated with $M = 1$, using the same square domain, with a circular piece removed. The boundary conditions along the circular curve are not provided. We tested three different sizes of circles as can be seen in Figure 6. The left column is the outcome of the algorithm inferred at the whole rectangle. The ground truth is on the right column and is as for the full shape. The error is depicted in the central column. It is easy to see that the error concentrates on the missing regions, and is extrapolated in a smooth fashion.

## 7 COMPARISON TO MATRIX-BASED METHODS

In this section we compare our algorithm with matrix-based methods where the continuous operator is discretized by finite differences scheme to form the 2D Laplacian matrix with spacing $h$. The approximation of the continuous operator, therefore, may yield significant numerical errors [Knyazev (2000)]. In addition, the incorporation of the boundary condition is not straightforward since the boundary conditions may affect the construction of the matrix approximation of the operator. We calculate the eigenpairs of the matrix by standard Matlab solver with $h = 0.03$. Quantitative results are shown in the last lines of tables 3 and 4. As can be seen, the proposed algorithm outperforms in the eigenvalue analysis (Table 4), while in the eigenfunction analysis the results are comparable. Next, we compare our results to the power method. This method is a well known iterative algorithm where given a diagonalizable matrix $A$ the algorithm finds the maximal eigenvalue and its corresponding eigenfunction, while the inverse power method finds the lowest one [Bronson et al. (2014)]. We adopted the inverse power method to our problem following [Bozorgnia (2016)]. Figure 7 shows the outcome of the inverse power method implemented in Matlab with $h = 0.03$ and initial eigenvalue $\lambda_0 = 1$ in two dimensions. On the left is the estimated eigenfunction which corresponds to the lowest eigenvalue ($m = 1$, $n = 1$). In the middle is the error $|u_{gt}(x, y) - u(x, y)|$ and in the right is the convergence plot of the eigenvalue which was converged to 2.02. In our ex-

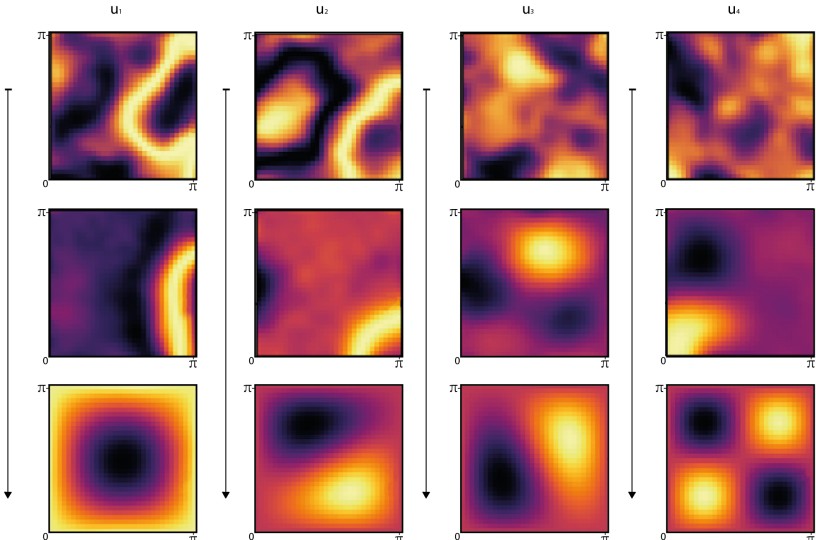

Figure 5: The solution to the 2D Laplace eigenvalue problem for $M = 4$. The Iterations $1, 100, 1000$ are shown from top to bottom, for each of the $4$ eigenfunctions.

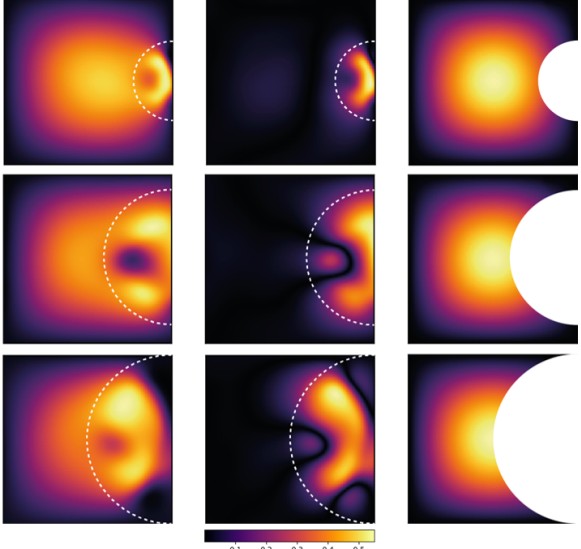

Figure 6: Free-form domain results for $M = 1$ excluding points in the right half circles. Left: The outcome of the proposed algorithm inferred in the full rectangle. The solution in the missing domain seems to be a smooth extrapolation. Middle: error from the full ground truth $|u - \text{full}(u_{\text{gt}})|$. The colorbar is referred only to this column. Right: Ground Truth. It is clear that the error is condensed into the missing domain, where the information is missing.

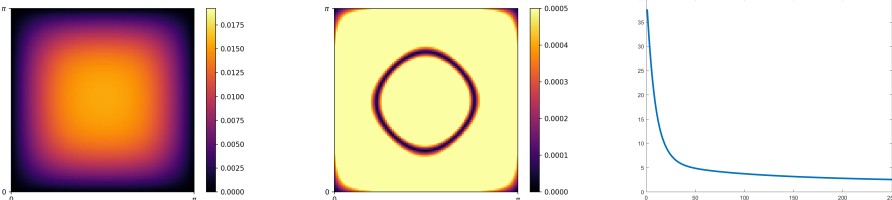

Figure 7: Inverse power method. Left: Approximated eigenfunction, Middle: error $|u_{gt} - u|$. Right: convergence of the eigenvalue. $\lambda = 2.02$, RMSE=8.76e$-$3, PSNR=20.57, SSIM=0.99

Table 3: Dimension= 2, Eigenfunction Error Analysis

| $M$ | Mean PSNR(per ef.) | Mean SSIM(per ef.) | Mean RMSE(per ef.) |
|---|---|---|---|
| 1 | 36.67 | 0.93 | 2.14e-4 |
| 2 | 31.66 | 0.89 | 1.33e-3 |
| 3 | 24.84 | 0.84 | 3.92e-3 |
| 4 | 18.97 | 0.52 | 1.52e-1 |
| 4 (Laplacian matrix) | 19.39 | 0.75 | 2.45e-1 |

periments, the algorithm seemed to be sensitive to discretization and initialization. Table 5 shows the performance of the proposed algorithm compared with the inverse power method. Better results are obtained in almost all quantitative measures. Furthermore, the inverse power method finds only a *single* eigenpair, while the proposed method outputs the $M$ smallest eigenpairs. Although it is possible to calculate more eigenpairs based on the previous one via orthogonality constraints, an undesired accumulated error may emerge.

# 8    IMPLEMENTATION DETAILS

Our network was constructed as a dense fully connected network with 5 hidden layers architecture, each with a varying number of neurons, from $26-100$. Our code was implemented in PyTorch. Each of our models was trained for 5000 iterations on NVIDIA QUADRO RTX-5000 Graphics card. In the 2D case all the networks were trained simultaneously. As for the activation function, since we are modeling smooth functions, we found that *ReLU* and its variants were less suitable both theoretically and experimentally. For the 1D experiments we used the $\tanh$ activation function, and in the 2D experiments we used the *SIREN* [Sitzmann et al. (2020)] activation function. The *SIREN* activation has been shown to excel in modeling complex signals, and their higher-order derivatives. We used an Adam optimizer [Kingma & Ba (2014)] with default parameters. A starting LR (Learning Rate) of $4e-3$ was used such that it was reduced every $100$ epochs by the factor of $0.7$ until the min LR reached $5e-5$.

We found that the weight initialization [Katanforoosh & Kunin (2019)] was important for convergence. In our 1D experiments a Gaussian initialization with 0.0 mean and 1.0 std was used. In the 2D experiments, we used the standard initialization proposed in [Sitzmann et al. (2020)]. Our hyperparameters were set as follows: $\alpha = 1e-1, \mu = 1e-1, \delta = 5e-1, \beta = 1.5, c = 1, \rho = 1e-8, \gamma_i = \frac{1}{i}, \nu = 2$. Our point set consisted of $45,000$ inner-points, and $1200$ boundary points. In the one-dimensional case, the boundary conditions had two actual points. In the two-dimensional case, points were taken along the boundary of the square. Instead of using an exact $\|\mathcal{T}u\|_\infty$, we used a relaxed approximation: $\frac{\mu}{K} \sum_{k \in \text{top}_K(|L_i|)} |L_k|$, with $K = 40$ for broader effect of the $L_\infty$ fidelity term.

Table 4: Dimension= 2, Eigenvalue Error Analysis

| $M$ | GT | Predicted | MAE | MRE | Rayleigh Quotient std |
|---|---|---|---|---|---|
| 1 | 2 | 2.01 | 0.0073 | 0.0037 | 0.0170 |
| 1 | 2 | 1.99 | 0.05 | 0.01 | 0.0432 |
|  | 5 | 4.91 |  |  |  |
| 3 | 2 | 1.94 | 0.07 | 0.019 | 0.0603 |
|  | 5 | 5.08 |  |  |  |
|  | 5 | 4.93 |  |  |  |
| 4 | 2 | 1.98 | 0.0490 | 0.0093 | 0.0234 |
|  | 5 | 4.93 |  |  |  |
|  | 5 | 4.96 |  |  |  |
|  | 8 | 7.91 |  |  |  |
| Laplacian Matrix |  |  |  |  |  |
| 4 | 2 | 1.96 | 0.1 | 0.02 |  |
|  | 5 | 4.90 |  |  |  |
|  | 5 | 4.90 |  |  |  |
|  | 8 | 7.84 |  |  |  |

Table 5: Comparison to the Inverse Power Method where $\lambda_{gt} = 2$

|  | $\lambda$ | PSNR | SSIM | RMSE |
|---|---|---|---|---|
| Inverse power method | 2.02 | 20.57 | **0.99** | 8.76e$-$3 |
| Proposed | **2.01** | **36.67** | 0.93 | **2.14**e$-$4 |

## 9 CONCLUSION

Interesting works have been recently published in the field of solving ODEs and PDEs with NNs [Raissi et al. (2017); Flamant et al. (2020); Bar & Sochen (2019); Sirignano & Spiliopoulos (2018)]. We propose a novel method to exploit these new innovations, along with specific constraints that exist in the mathematical nature of such problems, and shed some light on the well-known eigenvalue problem. Finding eigenpairs of both linear and non-linear differential operators is a problem that appears in many different areas in science and in many problems that arise in nature [Davydov (1976); Lpez-Blanco et al. (2013)]. Due to the possible complex nature of such operators, finding a robust method for solving this problem is challenging. Additionally, most such problems do not necessarily have a rectangular domain, and will therefore make the task more difficult across methods. Our method fits any free-form domain, and is able to alleviate these constraints. Furthermore, the proposed method can be used in principle on higher order, non-linear operators and on higher dimensional manifolds. When using the Rayleigh quotient to learn both the eigenfunction and its corresponding eigenvalue, the end-to-end learning process seems to assist the convergence, each acting as a constraint towards the other. When learning multiple eigenpairs at once, the orthogonality term eliminates degeneration. Empirically, it seems that these constraints also assist both the efs and the evs to converge towards more accurate solutions. This suggests that the orthogonality term serves as a regularizer on each eigenpair during the training process.

The training process produces *smooth* solutions, due to the fact that the networks are composed of linear layers and smooth activation functions (namely *tanh* and *SIREN*). Since these solutions are deterministic and parameterized by the learned weights, they are infinitely differentiable. The fact that the solution is given in an explicit form, enables an exact analytical differentiation. Rigorous analysis of the approximation error and its relation to the network architecture and design are under current study. Future research includes higher dimensions and non-linear operators.

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
