# OpenReview forum: "Deep Learning Solution of the Eigenvalue Problem for Differential Operators"
_ICLR.cc/2021/Conference — Reject_

### Official Review · AnonReviewer3 · 2020-10-26
**Lack important information: Reject**

**Rating:** 3
**Confidence:** 3

**Review:**

##########################################################################

Summary:

This paper proposes to use deep learning neural network to model eigenfunction while solving a generic eigenvalue problems. Since such problems is very generic this application of such framework are huge and the experiments show some promising results.

##########################################################################

Reasons for score:

Overall, I vote for rejecting (see cons for more details). The idea is very good, but many important information are missing. For example it is impossible to reproduce the results since we don't have any clue on the used networks.

##########################################################################

Pros:

1. This paper proposes a new framework for generic eigenvalue problem which seems promising.

2. The framework is able to deal with several eigenfunction and almost assure the orthogonality of the function.

3. This paper provides comprehensive experiments, that shows that such methods cam catch complex eigenfunction.

##########################################################################

Cons:

1. The notation is confusing on some parts. First please use <,> for scalar product instead of (,) since you have functions with two arguments. Second I don't w_u is a good notation for the weights , for example in equation 6 we don't know which weights it is. Perhaps using a function which gives the associated weights could help.

2.  When dealing with multiple eigenfunction, there is en additional term is the equation 6 for the orthogonality constraints. However it is not clear on how it is approximate, I assume that it is something similar to equation (3). Please clarify.

3. Again on the orthogonality constraint, the complexity of such constraint is huge. It is impossible to deal with many eigenfunction. Such limitation can drastically reduce the interest of the framework, I think there should be some discussion on this point.

4. On the implementation details, we don't have any clue on which kind of neural network are used. Are they just dense network?

5. Again on the implementation details, the loss function of equation 6 is highly non-convex and I assume that there is several way to try solving it. I don't think it is possible to deal with all the networks (the u) at once, so I figure there is a solving scheme (most likely alternate minimization) behind. Please clarify as it may answer to one of my previous remarks.

##########################################################################

Questions during rebuttal period:

Please address and clarify the cons above

#########################################################################

Some typos:

all pages: be careful with the citation, when citing a paper please add parenthesis

page 2: There is an extra parenthesis in equation 2

---

> ### Author Response · Authors · 2020-11-23
> **comments**
>
> We thank the reviewer for raising important issues. We thank for the encouraging points and answer the concerns:
>
> 1. We changed the notation of the scalar product for clarity. Additionally, we changed w_u to \theta_u as is common in the literature and clarified that these are the parameters of the network.
>
> 2. The implementation of the orthogonality constraint is as follows: we sum over all the points x_k the multiplication of all eigenfunctions u_i and u_j  such that i \neq j. We clarified it in the text (page 5).
>
>
> 3. Thanks for the important remark. Indeed, in the paper we used 4 networks in the 2D case. As mentioned in section 8 (Implementation details), the networks have only 5 fully connected layers with less than 100 neurons at each layer. Therefore, the complexity is not that huge. If we are interested in more eigenfunctions, we can change the architecture such that every network yields several eigenfunctions as we did in the 1D case.  This implementation is left for future work.
>
> 4. We clarified the description of the network architecture (5 layers dense network) in section 8 (Implementation details).
>
> 5. As the reviewer states, the optimization is non-convex. Yet, since the networks have 5 layers with at most 100 neurons per layer, we have at most 500 x 4 networks = 2000 parameters. This is a small number relative to standard networks like Resnet or VGG (few millions). The training procedure was straightforward with initialization and simultaneous training of all networks  (there is no need for alternate minimization nor fixing the eigenfunctions one after the other).
>
> 6. The citation format is according to the ICLR template, yet we added parenthesis.
>
> 7. Thank you very much, we corrected equation 2.

---

### Official Review · AnonReviewer4 · 2020-10-28
**I vote for reject.**

**Rating:** 4
**Confidence:** 5

**Review:**

The manuscript proposes a deep learning solver for the eigenvalue problem of differential self-adjoint operators. Specifically, the aim is to calculate M lowest eigenvalues and their corresponding eigenfunctions. This work is a natural follow up of the work by Bar and Sochen (2019) for solving PDE-based problems.  The unsupervised loss resembles the loss suggested in Bar and Sochen (2019), with addition of mainly two terms: (a) The Rayleigh Quotient term and (b) the orthogonality constraint.  In practice, the proposed framework is tested on the 1D and 2D Laplace operators on regular domains.

Pros.
1.	The authors address an important and practical problem
2.	The manuscript is self-contained, written very well and easy to follow
3.	The proposed framework has the potential to handle eigenvalue problem of self-adjoint differential operators over irregular domains.
4.	Mesh-free approach


Cons.

1.	Incremental novelty. I feel that that the proposed framework has a quite limited novelty as it naturally extends the work by Bar and Sochen (2019)
2.	Generalization.  Practically, the proposed framework is an optimization framework and not a training framework. In other words, each new instance requires re-training of the neural net, in case of modification of the coefficients of the differential operator  and /or the boundary conditions.  Moreover, in the 2D case, M networks are optimized simultaneously.
3.	I think that  it is worthwhile to refer to existing works, solving the eigenvalue problem by neural net.
4.	Limited experimental work.  The proposed framework has the potential to handle irregular domains. However, as far as I can see the experiments are done over regular domains for a special instance of differential operator, namely the Laplace operator.
5.	Comparison.   The comparison is made versus the inverse power method. I think that a comparison to a very relevant solver, LOBPCG, is missing.

---

> ### Author Response · Authors · 2020-11-23
> **Comments**
>
>
> We thank the reviewer for raising important issues.
>
> 1. The solution of PDEs via neural networks has emerged in recent years and is still very challenging for every specific application. The idea of parameterization the solution via a neural network is a very general framework where each equation or problem we address requires the proper formulation, prior knowledge regularization, definition of the boundary conditions, geometry, selection of architecture,  hyperparameter and so forth. Designing a network with arbitrary coefficients and boundary conditions in an unsupervised fashion is of a great challenge and opens new directions for future research.
> In this work we address the problem of finding the M-lowest eigenpairs of differential operators with arbitrary geometry and boundary conditions.
> Saying all this we would like to draw your attention that in previous works a one given PDE was solved. One major contribution of this work is the possibility to handle a family of PDEs  (one for each eigenvalue and finding the eigenvalues themselves) in one network that solves these multiple PDEs together.
> Finding the M-lowest eigenpairs was addressed in the literature as the power or inverse power methods, Rayleigh–Ritz methods, LOBPCG, and more. There are few fundamental contributions of the proposed framework:
>
> a) The above methods find eigenpairs of matrices. It means that some discretization was applied to approximate the continuous operator. This in turn yields an approximation error.
>
> b) Moreover, the power method for instance calculates only ONE eigenpair. It is possible to calculate more eigenpairs based on the previous one by orthogonality constraints. Nevertheless, an undesired accumulated error may emerge.
>
> c) Third, the boundary conditions should be integrated in the discretized operator. The modification to this matrix is not straightforward and requires special schemes especially for Neuman boundary conditions.
> In contrast to these methods, the proposed algorithm finds the eigenpairs simultaneously in the continuous domain where the boundary conditions are easily integrated in the loss function. Moreover, our framework can easily address non-standard geometries since we work with point cloud instead of a mesh. We used standard rectangular domain where analytic solutions exist to compare our method to the ground truth. We added, in this revised version,  an example of a rectangle cut by half a circle with 3 different sizes (Figure 6) and demonstrate the ease that our proposed method deals with more complex geometries.
> We clarified these contributions in the revised version.
>
> 2) We added references to previous methods that address the eigenvalue problem using neural networks. These methods are based on variational Monte Carlo (VMC) and diffusion Monte Carlo (DMC) methods. VMC relies on leveraging physical knowledge to propose an ansatz of the eigenfunction and incorporates the essential physics.
> Recently, Han et al  formulated the eigenvalue problem by the stochastic backward equation using the DMC method. In this method, the loss function optimized the eigenvalue, eigenfunction and the scaled gradient of the eigenfunction. The loss function consists of L2 norm of two fidelity terms with additional normalization.  The algorithm yields the first eigenpair with an optional second eigenpair given some mild prior estimate of the eigenvalue.
> In the suggested work, we formulate the eigenvalue problem in a direct setting with flexible number of eigenpairs. Additionally, we use L infinity norms for fidelity and boundary condition terms to accomplish a strong (pointwise) solution.
>
> 3) Thank you very much for this comment. Indeed, we added an experiment where we cut the rectangle with half a circle with 3 different sizes (Figure 6). By that we demonstrate our claim.
>
> 4) The LOBPCG method approximates the continuous operator by finite differences approximation unlike the proposed method where we work directly in the continuous space. As cited in the original paper[1]:
> "Typical properties of mesh eigenproblems are well known; see, e.g., [22].
> We just want to highlight that the desired eigenpairs of the matrix pencil B − µA are
> rarely needed with high accuracy as the pencil itself is just an approximation of the original
> continuous problem and the approximation error may not be small in practice.”
> We built the 2D Laplacian matrix and calculated the eigenvalues using a Matlab solver.  The comparison is shown in the last lines of tables 3, 4.
>
> [1] A. Knyazev, "Toward The Optimal Preconditioned Eigensolver: Locally Optimal Block Preconditioned Conjugate Gradient Method",  SIAM Journal on Scientific Computing 23(2), 2000

---

### Official Review · AnonReviewer1 · 2020-10-29

**Rating:** 4
**Confidence:** 2

**Review:**

The paper proposes a framework to estimate the eigenfunctions and eigenvalues of (non-linear) differential operators. The main idea is to parameterize the eigenfunction with a neural network and minimize a loss function which enforces the definition of eigenfunctions along with auxiliary terms enforcing boundary conditions, smoothness and getting rid of de-generate solutions.

My main concern is the lack of novelty/understanding what the contribution is. Going through the existing literature (section 2.1 in related work), it seems that the standard way to solve a differential equation using deep networks is to parameterize the solution using a deep network and optimize a loss which captures the physical relation (that is the PDE itself) along with auxiliary terms for smoothness, boundary condition etc. In that sense, it is not clear to me what the contribution of the paper is. The main loss in the paper is definition of eigenfunction since that is the problem they're interested in and in the literature for solving PDEs, the main loss is the definition of the PDE itself. The authors should put their contribution in context with existing literature and highlight the changes they've made.

In addition, it's unclear to me why we are particularly interested in the lowest M-eigenpairs. Is there a reason to concentrate on the lowest eigenvalues instead of the highest? I don't think this is motivated in the paper. Further, section 7 says " Furthermore, the inverse power method finds only a single eigenpair, while the proposed method outputs the M smallest eigenpairs.". Can we not find the smallest pair and run the method again with a constraint that the eigenfunction should be orthogonal to the one we already found to find the next smallest eigenfunction? (similar to what we do in power method?)

Finally, the experiments only concentrate on a Laplacian operator. In order to claim that is general and works across multiple operators, I believe that the results should be demonstrated  on multiple differential operators (of varying complexity levels). Unfortunately, I do not know enough about differential operators to suggest what these test cases should be.

---

> ### Author Response · Authors · 2020-11-23
> **Comments**
>
> We thank the reviewer for the important comments.
>
> 1. The solution of PDEs via neural networks has emerged in recent years and is still very challenging for every specific application. The idea of parameterization the solution via a neural network is a very general framework where each equation or problem we address requires the proper formulation, prior knowledge regularization, definition of the boundary conditions, geometry, selection of architecture, hyperparameter and so forth. It is analogous to use general mathematical tools like linear algebra or calculus of variations where every problem must be carefully formalized.
> Saying all this we would like to draw your attention that in previous works a one given PDE was solved. One major contribution of this work is the possibility to handle a family of PDEs  (one for each eigenvalue and finding the eigenvalues themselves) in one network that solves these multiple PDEs together.
> In this work we address the problem of finding the M-lowest eigenpairs of differential operators with arbitrary (regular) geometry and boundary conditions. Finding the M-lowest eigenpairs was addressed in the literature as the power or inverse power methods, Rayleigh–Ritz methods, LOBPCG, and more. There are few fundamental contributions of the proposed framework:
>
> a) The above methods find eigenpairs of matrices. It means that some discretization is applied to approximate the continuous operator and in turn yields an approximation error.
>
> b) Moreover, the power method for instance calculates only ONE eigenpair. It is possible to calculate more eigenpairs based on the previous one by orthogonality constraints. Nevertheless, an undesired accumulated error may emerge.
>
> c) Third, the boundary conditions should be integrated in the discretized operator. The modification to this matrix is not straightforward and requires special schemes for Neuman and Dirichlet boundary conditions.
>
> In contrast to these methods, the proposed algorithm finds the eigenpairs simultaneously in the continuous domain where the boundary conditions are easily and naturally integrated in the loss function. In turn, we do not have the approximation errors. Moreover, our framework can easily cope with non-standard geometries since we work with point cloud instead of a mesh. We added an example of a rectangle cut by half a circle of three different sizes and demonstrate this capability. We also clarified these contributions in the revised version.
>
> 2. Shape correspondence/classification is a key problem in computer vision, computer graphics and related fields with a broad range of applications, including texture or deformation transfer and statistical shape analysis. The M-smallest eigenvalues of the Laplace-Beltrami operator are fundamental features for comparing geometric objects such as 3D shapes, images or point clouds via the functional maps method. We added this motivation to the revised paper. Note that in quantum mechanics the smallest eigenvalues and eigenfunction of the Hamiltonian are of great physical significance.
>
> 3. It is true that one can find the smallest eigenpair by the power method and then find another one such that the next eigenfunction is orthogonal to the previous. Nevertheless, the errors are accumulated during the iterative process which may yield undesirable results. In the proposed method we simultaneously calculate all the M eigenpairs and make sure the orthogonality constraint is satisfied.
>
> 4. Indeed, in future work we intend to implement the algorithm on other differential operators as well. Due to the complexity of this task it is beyond the scope of the ICLR conference.

---

### Official Review · AnonReviewer2 · 2020-10-30
**Solving for eigenvalues via unsupervised ANN regression**

**Rating:** 9
**Confidence:** 2

**Review:**

The authors frame the decomposition of the Laplacian equation as an unsupervised regression problem that is using a 5-level (and fully connected?) neural network as regression function.  A cost function to be used in the optimization is proposed that is expanded to eigendecomposition problems of increasing complexity. A comparison with a classical method indicates that the proposed approach is comparable to or better than the former for the given task.

I like the overall approach and the idea of framing the solution of the eigenvalue decomposition as an ANN regression problem, and I would see that this is of interest to ICLR. From an application perspective, the authors refer to computer vision tasks, I would be interested in a somewhat deeper discussion of constraints of the approach, for example:

As both methods are somewhat on par in terms of various performance indicators I was wondering whether the proposed approach is faster?

In real world problem critical information is contained in the boundary conditions. Could the authors discuss pros and cons of their approach with respect to them? And, again, compare to existing methods?

---

> ### Author Response · Authors · 2020-11-23
> **Comments**
>
> We thank the reviewer for the encouraging feedback.
>
> 1. Yes, we used a dense (fully connected) networks and clarified it in the text (Implementation details section).
> 2. As for the performance, in all our experiments the running time was at most 5 minutes (with GPU) since the networks are not very deep and the number of trainable parameters was at most 2000. The inverse power method took approximately 4 minutes (without GPU).  Note that in classic numerical approaches one has to discretize first the problem and to represent it as an eigen problem for a matrix. This in itself is prone to numerical errors while in our approach we avoid this step.
> 3. Regarding the limitation of the method: we still need to modify the scalability of the method in terms of the number of eigenfunctions. We consider a redesign of the architecture (especially in 2D or more) such that every network yields multiple eigenfunctions. This in turn would reduce the number of networks we need to train. Moreover, we need to investigate higher dimensions and non-linear operator as well.
> 4. In the present formulation, the network must be trained again if the boundary conditions are changed. This is an interesting direction for future research to design a system which will only make minor adaptation to this change.
> The incorporation of the boundary condition in existing methods, in particular LOBPCG, power method and other matrix-based operators is not straightforward since the boundary conditions may affect the construction of the matrix approximation of the operator.
> We explained these points in the text.

---

### Author Response · Authors · 2020-11-23
**General comments**

We thank the reviewers for the constructive and important comments. We did our best effort to answer all the concerns. We added references, more discussions, and demonstrate the method with non-standard shapes.
For convenience, the changes in the revised paper are marked in blue.

---

### Decision · Program_Chairs · 2021-01-07
**Final Decision**

**Decision:**

Reject

**Comment:**

The paper addresses the problem of solving for the eigenpairs of a self-adjoint differential operator. This problem, of course, is classical; the main innovations here are
a) the use a parametric form of the (pointwise) solution using a (shallow) neural network so as to avoid discretization, and
b) obtaining multiple eigenpairs simultaneously as outputs.
The techniques themselves (i.e., the architecture, the loss function, and the training procedure) are fairly standard.

There was some variance in the review scores. The reviewers appreciated the importance of the problem and the direction adopted by the authors. However, concerns were raised regarding: the limitations of the experimental evaluation; a lack of sufficient distinction from prior work; *very* limited comparisons with prior approaches; and the limited demonstration of generalizability. I agree with all these criticisms, and therefore vote to reject.